# Cellular and Molecular Activities of IP6 in Disease Prevention and Therapy

**DOI:** 10.3390/biom13060972

**Published:** 2023-06-10

**Authors:** Lowell Dilworth, Dewayne Stennett, Felix Omoruyi

**Affiliations:** 1Department of Pathology, The University of the West Indies, Mona Campus, Kingston 7, Jamaica; 2The Transitional Year Programme, University of Toronto, Toronto, ON M5S 2E8, Canada; 3Department of Life Sciences, Texas A&M University, Corpus Christi, TX 78412, USA; 4Department of Health Sciences, Texas A&M University, Corpus Christi, TX 78412, USA

**Keywords:** IP6, phytic acid, disease, molecular, anti-oxidant, anti-inflammatory, diabetes, lipid

## Abstract

IP6 (phytic acid) is a naturally occurring compound in plant seeds and grains. It is a poly-phosphorylated inositol derivative that has been shown to exhibit many biological activities that accrue benefits in health and diseases (cancer, diabetes, renal lithiasis, cardiovascular diseases, etc.). IP6 has been shown to have several cellular and molecular activities associated with its potential role in disease prevention. These activities include anti-oxidant properties, chelation of metal ions, inhibition of inflammation, modulation of cell signaling pathways, and modulation of the activities of enzymes and hormones that are involved in carbohydrate and lipid metabolism. Studies have shown that IP6 has anti-oxidant properties and can scavenge free radicals known to cause cellular damage and contribute to the development of chronic diseases such as cancers and cardiovascular diseases, as well as diabetes mellitus. It has also been shown to possess anti-inflammatory properties that may modulate immune responses geared towards the prevention of inflammatory conditions. Moreover, IP6 exhibits anti-cancer properties through the induction of cell cycle arrest, promoting apoptosis and inhibiting cancer cell growth. Additionally, it has been shown to have anti-mutagenic properties, which reduce the risk of malignancies by preventing DNA damage and mutations. IP6 has also been reported to have a potential role in bone health. It inhibits bone resorption and promotes bone formation, which may help in the prevention of bone diseases such as osteoporosis. Overall, IP6’s cellular and molecular activities make it a promising candidate for disease prevention. As reported in many studies, its anti-inflammatory, anti-oxidant, and anti-cancer properties support its inclusion as a dietary supplement that may protect against the development of chronic diseases. However, further studies are needed to understand the mechanisms of action of this dynamic molecule and its derivatives and determine the optimal doses and appropriate delivery methods for effective therapeutic use.

## 1. Introduction

IP6 (phytic acid) is the primary storage form of phosphorus in many food crops and the longevity of seeds up to 400 years has been ascribed to the presence of IP6 [1,2,3]. IP6 is a poly-phosphorylated inositol carbohydrate (C_6_H_18_O_24_P_6_) (Figure 1) that has been shown to exhibit many biological activities that include anti-neoplastic properties [4,5], inhibition of metastatic tumors [6,7], as well as hypoglycemic and hypolipidemic activities [8,9,10,11]. Due to its ability to chelate metal ions and inhibit their absorption from the intestinal tract, IP6 has been generally considered an anti-nutrient [12,13], in addition to its capacity to bind with proteins and starch [14].

Still, these attributes of IP6 could also explain its beneficial effects, some of which will be explored further [15,16]. IP6 influences cellular functions such as DNA repair, chromatin remodeling, endocytosis, and nuclear messenger RNA export [17]. Shamsuddin et al. [18] reported that IP6 in cells de-phosphorylates to form lower inositol phosphates (IP1–IP5). IP6 and IP5 have the most significant anti-nutritional effects compared with the smaller molecules (IP4, IP3, IP2, IP1) [19] because of their higher capacity to complex with inorganic cations. These smaller molecules of lower inositol phosphates function as intra-cellular messengers in cellular signaling and regulate cell growth and differentiation [20]. Some researchers have postulated that IP6 may be transported into the gastrointestinal epithelial cells intact, where it is de-phosphorylated [21], as the intra-cellular concentration of lower inositol phosphates has been shown to increase with IP6 supplementation [22,23]. Grases et al. [23] also suggested that maintaining the appropriate levels of IP6 in humans requires the consumption of IP6 supplements when the diet is poor in phytate. Verbsky et al. [24] also proposed the human pathway for IP6 production to involve the isomerization of Ins(1,4,5)P3 to Ins(1,3,4)P3 and the sequential phosphorylation of Ins(1,3,4)P3 to Ins(1,3,4,6)P4 by the IP3 5/6 kinase. Then, the conversion of Ins(1,3,4,6)P4 to Ins(1,3,4,5,6)P5 by an IP4 5-kinase and the subsequent production of IP6 from IP5 by IP5 2-kinase. Notable roles of IP6 in human health are linked to its anti-oxidant and anti-carcinogenic activities as well as its functioning as a regulator of gene expression. IP3 containing 1,2,3-triphosphate and 1,2,6-triphosphate are iron-binding anti-oxidants while also possessing anti-inflammatory/secretory properties, respectively [25,26,27]. On the other hand, IP4 containing 1,2,3,6 tetraphosphate is moderately effective in opening calcium channels [28,29]. Shamsuddin et al. [7] demonstrated that the combination of IP6 and inositol in the appropriate ratio produces inositol 1,4,5-trisphosphate (Ins (1,4,5) P3 or IP3) signaling molecules, recognized as essential cellular regulators. We also reported that combined inositol and IP6 supplementation might be helpful in the management of type 2 diabetes mellitus and related metabolic disorders by regulating some aspects of lipid and carbohydrate metabolism [30]. IP6 is either absorbed from the stomach and small intestines or broken down in the large intestines of monogastric animals, such as humans, by gut microbes to lower inositol phosphates [31]. The safety of IP6 consumption, as well as how efficiently it is absorbed from the digestive tract, is still being debated among researchers [28,29,30]. Ullah and Shamsuddin [32] reported the non-toxicity of IP6, thus deeming it safe for consumption, even with long-term use. Vucenik and Shamsuddin [4] also showed that IP6 is safe and devoid of toxicity in vivo. However, Zhou and Erdman [17] reported that excessive IP6 consumption might be associated with negative mineral balance because of its insoluble complex formation with metals. Hence, chronic use must be carried out with caution as its ability to chelate cations may lead to mineral deficiencies in nutrient-poor diets [33]. IP6 is also believed to form a complex with proteins that may lead to the modulation of digestive enzymes in the gastrointestinal tract, which could have an overall impact on nutrient absorption [19,34]. However, IP6’s ability to reduce intestinal mucosa amylase and Na^+^/K^+^ ATPase activities may accrue some benefits in managing diabetes by decreasing intestinal carbohydrate digestion and absorption and reducing blood glucose levels [8,35,36]. IP6 also modulates essential multi-cellular functions, including cell proliferation, signal transduction, and differentiation [37,38]. IP6 exerts its differentiating activity via de-phosphorylated derivatives (IP4, IP5), allowing cells to return to normal proliferative behavior [39]. Developing new blood vessels is essential for the growth of solid tumors. Hence, the anti-angiogenic property of IP6 may accrue some benefits in this regard [40]. Bozsik et al. [41] reported the up-regulation of some genes involved in this process, especially upon five mM of IP6 treatment. There are also reports highlighting its beneficial properties on bone health. Overall, studies involving animals and humans agree that increased oral administration of IP6 leads to increased urinary excretion of total IPs and excretion of mixtures of multiple different InsPs [42]. The in vivo synthesis of IP6 may only meet the body’s needs with dietary supplementation. Grases et al. [23] reported a maximum excretion level that cannot be exceeded by ingesting a higher amount of IP6. They suggested that the urine IP6 level can be a marker to monitor its level in the body because IP6 levels in urine correlate to blood levels. Thus, a suitable adjustment of dietary IP6 level is needed to allow for the beneficial effects and minimize the negative impact associated with IP6 [43]. In the following brief review, we will discuss the broad applicability of IP6 from nutraceutical and pharmaceutical perspectives.

## 2. Anti-Diabetic Activity of IP6

Diabetes mellitus is a metabolic disorder involving carbohydrate metabolism alterations with associated hyperglycemia. Hyperglycemia may be due to a relative or absolute deficiency of insulin and/or insulin resistance associated with disruptions in carbohydrate, lipid, and protein metabolism. The conventional treatment of diabetes includes diet modification, increased physical activity, oral hypoglycemic drugs, and insulin therapy. The treatment cost of diabetes is high due to the associated complications and medications used for its treatment have several side effects. Globally, medicinal plants are increasingly being utilized as an alternative treatment option [44], yet the primary challenge lies in the requirement for greater scientific backing. IP6 is present in almost all plant and mammalian cells as an organic phosphorus compound and is a reservoir of phosphorus in grains and oilseeds [45]. Food sources high in fiber, such as wheat bran and flaxseed, are also high in phytates [46,47]. Phosphate and inositol are the two constituents of IP6. Studies have highlighted the efficacy of IP6 in helping to reduce the incidence of diabetes mellitus while improving outcomes in patients afflicted with the disease. The mineral-chelating properties have been used to deter Fe^3+^ metal-catalyzed protein glycation end products, which contribute to diabetic complications, including nephropathy, retinopathy, and neuropathy [48]. This mineral-chelating property is also purported to be responsible for a reduction in the glycemic index value.

By chelating Ca^2+^ ions, a co-factor of α-amylase, IP6 reduces the rate of starch digestion. Studies have shown that, along with inhibiting α-amylase activities, IP6 also inhibits α-glucosidase activities in a dose-dependent manner [10]. IP6 affects carbohydrate metabolism by forming complexes directly to starch or to the proteins associated with it [14,49], reducing its digestibility, bioavailability, and glycemic index value. Dilworth et al. [8] (Table 1) also reported lowering blood glucose levels through IP6 supplementation. Omoruyi et al. [50] (Table 1) reported a lowering of blood glucose in diabetic rats administered a phytate supplement, which they also attributed to decreased intestinal amylase activity.

The association between type 2 diabetes and a low-grade inflammatory state, which involves the activation of innate immunity [65], has also been established. Therefore, it is important to identify molecules that target inflammatory mechanisms to improve the diabetic state. IP6 has been reported to modulate immune system functions by promoting natural killer (NK) cell activity, regulating neutrophils, and reducing pro-inflammatory cytokine expression [66]. IP6 has exhibited anti-inflammatory activity [52,54] (Table 1), which could be mediated through its anti-oxidant properties and modulatory effects on macrophages. Numerous studies have established the involvement of redox imbalance in developing diabetes complications. There is growing evidence that IP6 possesses anti-oxidant properties that may inhibit lipid peroxidation [67] and hydroxyl radical formation [68], which may also accrue some benefits to diabetic individuals. IP6′s ability to lower blood glucose levels, improve insulin sensitivity, reduce carbohydrate digestion and subsequent absorption in the gut and inhibition of protein glycation makes it a promising natural therapeutic agent for the management of diabetes and the prevention of complications associated with the disease.

Vucinek and Shamsuddin [37] (Table 1) suggested that combining IP6 and inositol might be more effective in maintaining human health than each component alone. We have also previously demonstrated that combined IP6 and inositol may effectively manage diabetes mellitus in animal models of type 2 diabetes [30,35,36,51] (Table 1). We noted that combined inositol and phytic supplementation in diabetic rats lowered non-fasting blood glucose levels. We also attributed it to inhibiting intestinal amylase activity with the associated step-down of carbohydrate digestion in the gut [47]. One of the mechanisms of phytic acid in lowering blood glucose levels may be via reduced carbohydrate digestion and subsequent absorption in the gut. Foster et al. [30] reported that the hypoglycemic activity of combined inositol and IP6 might also be partly due to their observed decreasing pattern in Na^+^/K^+^ ATPase activity that may slow down the absorption of glucose across the intestinal mucosa. In one of our studies, we highlighted the effect of inositol and IP6 combination on the serum leptin concentration in type 2 diabetic rats. We hypothesized in that study that the significant increase in serum leptin levels might increase hepatic insulin sensitivity. The combination treatment significantly reduced insulin resistance towards normal levels [30]. We then theorized that combined inositol and IP6 acting synergistically may be beneficial to people with diabetes by reducing the risks associated with obesity and other downstream pathological complications [35]. Kim et al. [53] reported that IP6 and myo-inositol increased adipogenesis (insulin-stimulated glucose uptake in adipocytes via insulin signaling activation) and inhibited basal lipolysis in mature adipocytes (Table 1). Sustained hyperglycemia is associated with the glycation of vital biomolecules and the formation of advanced glycation end-products that contribute to diabetic complications. However, the daily consumption of IP6 inhibits protein glycation in type 2 diabetic patients, resulting in decreased diabetic complications [48] (Table 1). Vucenik et al. [69] noted that certain parameters of some tumor models may remain unchanged or worsen when treated with inositol or IP6 alone. However, they observed that combining inositol and IP6 yielded significantly better results in different cancer types than using either one alone. As a result, they concluded that, in clinical settings, it is not advisable to use inositol or IP6 alone as they are not optimally effective, instead, they work best when used together.

## 3. Anti-Inflammatory Activity of IP6

Inflammation is an immediate defense of the host to eliminate harmful stimuli [70,71,72,73]. Cytokines that promote inflammatory response are called “proinflammatory cytokines.” On the other hand, “anti-inflammatory cytokines” down-regulate the inflammatory response by suppressing the activity of pro-inflammatory cytokines [74]. However, the immune system’s role is to balance the activity of pro-inflammatory and anti-inflammatory mediators [75]. The over-expression of the immune response could result in excessive reactive oxygen species that can cause damage to DNA and cellular membranes [75]. IP6, a natural compound found in plants, is believed to have potential health benefits, including the ability to reduce inflammation in mammalian cells. One of the genes related to inflammation and cancer encodes inducible nitric oxide synthase (iNOS). iNOS catalyzes the synthesis of nitric oxide (NO) from the oxidative deamination of L-arginine [75]. The two isoforms of nitric oxide synthase constitutively expressed are endothelial (eNOS) and neural nitric oxide synthase (nNOS). The inducible isoform is expressed in inflammation [76] and is regulated at the transcriptional level by cytokines such as IFN-γ, TNF-α, IL-1β, and IL-6 and by bacterial lipopolysaccharides (LPS) [77]. Feghali and Wright [78] showed that acute inflammation is associated with the production of cytokines interleukin (IL)-1, IL-6, and tumor necrosis factor-α (TNF-α).

In contrast, chronic inflammation includes cytokines that are involved in humoral such as IL-3, IL-4, IL-5, IL-6, IL-7, IL-9, IL-10, IL-13, and transforming growth factor-b (TGF-b) and cellular contributors such as IL-1, IL2, IL-3, IL-4, IL-7, IL-9, IL-10, IL-12, interferons (IFNs), IFN-g inducing factor (IGIF), TGF-b, and TNF-a and -b [78]. Sultani et al. [79] demonstrated that the cytokines IL-1β, IL-6, and TNF-α are strong promoters of inflammation, while IL-4 and IL-10 are anti-inflammatory. IL-4 targets IL-1β to suppress inflammation, while IL-10 targets both IL-6 and TNF-α to block and regulate inflammation. The liver is a central regulator of inflammation by its ability to secrete several proteins that regulate the systemic inflammatory response and is the target of inflammatory reactions [80,81]. Because of its central location in metabolism regulation and response to exogenous stimuli, both physiological and pathological, every form of liver disease is accompanied by some degree of inflammation. Pro-inflammatory factors are key regulators that promote the continuous development of non-alcoholic fatty liver disease (NAFLD). At the same time, IL-1β and IL-6 stimulate the formation of hepatic collagen and other substances associated with liver damage by promoting liver fibrosis [82]. Tumor necrosis factor-α induces liver necrosis by increasing mitochondria reactive oxygen species with associated lipid peroxidation [83]. However, Omoruyi et al. [50] noted that IP6 supplementation significantly increased the serum level of IL-1β and a rising trend in IL-6 serum levels in diabetic rats and concluded that the significant elevation of IL-1β, a pro-inflammatory cytokine, by phytate supplementation, might contribute to the hypertrophy of liver tissue with subsequent leakage of alanine aminotransferase and alkaline phosphatase from the liver into the serum. They, however, concluded that the observed adverse effect on the liver might be due to the combined effect of streptozotocin-induced diabetes and IP6 supplementation. On the contrary, IP6 was recently reported to lower the production of pro-inflammatory mediators [84], which may support the beneficial potential of IP6 in the management of diseases associated with pro-inflammatory mediators’ generation.

The NF-kB signaling pathway regulates the inflammatory response by promoting the transcription of the inflammatory factors [85]. Hence, the effective treatment for lowering disease inflammation should also be geared towards targeting the NF-kB. Without inflammation, NF-kB P65 in the cytoplasm binds tightly to IκBs (inhibitors of NF-κB). This complex is disintegrated by inflammatory stimulators such as TNF-α, IL-1, or LPS through the phosphorylation of IκB proteins, with the subsequent release of the active form of NF-κB that is then translocated to the nucleus where it up-regulates gene expression [86]. In the study by Ran et al. [84], they observed LPS-induced phosphorylation of IκBα and NF-κB P65 in mice and mMECs and reported that IP6 inhibited their phosphorylation. They also noted that IP6 inhibited LPS-induced phosphorylation of the protein kinases Akt and p38 but did not affect the phosphorylation of c-Jun N-terminal kinase (JNK) or extracellular signal-regulated kinase 1/2 in vitro and in vivo. Thus, IP6 may not be a universal inhibitor of the phosphorylation of targets in the mitogen-activated protein kinases (MAPK) signaling pathway that promotes pro-inflammatory factor expression [87,88]. The recent report by Ran et al. [84] indicated that IP6 supplementation significantly lowered pro-inflammatory factors in the liver of high-fat diet-fed mice and the phosphorylation of NF-κB p65 in high-fat diet-fed mice liver; however, it supported its potential use in the management of inflammation associated with diabetes and the curtailment of complications attributable to the disease. Overall, IP6 supplementation possesses anti-inflammatory properties and may have a potential role in mitigating inflammatory responses in various diseases, including diabetes, via its ability to reduce production of pro-inflammatory cytokines and inhibit the activation of inflammatory pathways in the body.

## 4. Anti-Oxidant Activity of IP6

Under normal physiological conditions, there is a balance between oxidants and anti-oxidants to prevent tissue damage associated with free radical toxicity. However, in oxidative stress, this balance is disturbed, resulting in the excessive generation of free radical species [89,90,91,92,93]. Free radicals can be formed from di-oxygen molecules containing unpaired electrons [94,95,96,97]. It is precisely because of the unpaired electrons that oxygen molecules react easily with other molecules [98,99,100,101]. This results in chemical chain reactions that can have beneficial or harmful consequences [102,103,104]. Increases in the concentrations of ROS are associated with lipid peroxidation, protein oxidation, DNA damage, enzyme inhibition, and the activation of programmed cell death pathways [105]. The severity of oxidative stress induced by reactive oxygen species (ROS) is concentration-dependent and based on the type of cells involved and exposure duration. While low doses of ROS are mitogenic and promote cell proliferation, intermediate levels temporarily or permanently result in growth arrest and high levels result in cell apoptosis or necrosis [106]. IP6 has been shown to promote apoptosis and inhibit cellular growth in several types of cancers [107,108]. Many of the beneficial effects of IP6 have been attributed to its anti-oxidant properties and its ability to ameliorate oxidative stress by metal ion chelation [34,109]. Kato et al. [110], however, stated that the amelioration of oxidative stress does not sufficiently explain IP6′s ability to induce cell death. They showed that IP6 induced necrotic cell death by promoting the amplification of ROS generated by NADPH oxidase. 

The anti-oxidant system is composed of enzymatic and non-enzymatic anti-oxidants. The enzyme-based anti-oxidant system includes superoxide dismutase (SOD), catalase (CAT), and peroxidases. Superoxide dismutase catalyzes the conversion of highly reactive superoxide radicals into hydrogen peroxide and then to water and oxygen by CAT. Like CAT, glutathione peroxidase (GPx) converts hydrogen peroxide into non-toxic molecules, thus protecting the cell from oxidative damage. This reaction involves the oxidation of reduced glutathione (GSH) and results in the formation of oxidized glutathione (GSSG), which can then be reduced back to GSH by glutathione reductase [105]. The ratio of GSH/GSSG is an indicator of intra-cellular oxidative status. IP6′s involvement in protecting cells from oxidative stress through modulating CAT, GPx, and SOD activities while reducing lipid peroxidation has been reported [55,111]. Lee et al. [55] (Table 1) showed that IP6 supplementation significantly reduced CAT activity in the liver of rats treated with a carcinogen by scavenging the origin of free radicals. Foster et al. [51] showed that a combination of IP6 and inositol effectively up-regulated hepatic SOD and CAT activities in diabetic rats. The non-enzymatic anti-oxidants include GSH, bilirubin, uric acid, vitamin C, carotenoids, and some other phenolic compounds [105,112]. Like IP6, many of these anti-oxidant molecules can be supplemented through daily diets. IP6 has been shown to modulate the levels of endogenous non-enzymatic anti-oxidants such as GSH. Diets of diabetic rats supplemented with a combination of IP6 and inositol resulted in significant increases in hepatic GSH and thus may be effective at preventing increases in lipid peroxidation in diabetic complications [111]. All the effects of IP6 outlined above are thought to contribute to the overall anti-oxidant characteristics of the molecule. IP6 can scavenge free radicals, inhibit lipid peroxidation, and reduce oxidative stress in various tissues. It may prevent the development of some diseases and accrue benefits in managing diseases.

## 5. Anti-Cancer Activity of IP6

A plethora of studies have investigated the anti-cancer properties of IP6 [56,57,58] (Table 1). IP6 and its derivatives, such as IP5, IP4, and IP3, have been shown to possess anti-cancer activities by modulating biological processes implicated in various cancers, including inflammation, apoptosis, angiogenesis, proliferation, cell signaling, and gene expression [31,113,114]. Some studies have highlighted the potential cytostatic and cytotoxic properties of IP6 in malignant cells [59,115,116] (Table 1). Other reported anti-cancer effects of IP6 include boosting immunity and anti-oxidant activities [60] (Table 1).

The overexpression of iNOS is linked to chronic inflammatory diseases and different cancers [117]. The anti-cancer and anti-inflammatory activities of IP6 may be through the suppression of the gene expression that encodes iNOS [118]. IP6 decreases the expression of tumor necrosis factor-α (TNF-α), activates caspase-3 and p53, and inhibits the activation of MAPKs [66,119,120]. The decrease in the expression of TNF-α is also accompanied by modulation of the expression of its receptors, TNFRI and TNFRII. Much of the biological effects of TNF-α are mediated through these receptors. The expression of TNFRI, which is involved in programmed cell death, is increased, while the expression of TNFRII, which promotes some undesirable effects, is decreased [52].

Modulation of intra-cellular signaling cascades that include phosphatidylinositol-3 kinase (PI3K), protein kinase C (PKC), activator protein-1(AP-1), and nuclear factor-kappaB (NFκB) are linked to the anti-cancer activity of IP6 [38,121]. Persistent activation of the NFκB pathway has been said to promote processes involved in cancer development and treatment resistance [122]. IP6 has been shown to inhibit NFκB activation in cervical cancer HeLa cells and prostate cancer DU-145 cells [120,123]. Activator protein-1, a major transcription factor whose activation contributes to tumorigenesis, is inhibited by IP6 in human prostate carcinoma PC-3 cells [124]. The activity of PKC delta, a major PKC isoform, has been shown to be up-regulated in IP-6-induced apoptosis in MCF-7 human breast cancer cells treated with 2 mM IP6 for 24 h [61] (Table 1). PI3K, whose activation promotes the survival, proliferation, and angiogenesis in human prostate cancer cells, is inhibited by IP6, thereby protecting cells from carcinogenesis and preventing tumor angiogenesis [108]. As an essential signaling molecule, it acts in conjunction with signaling proteins to suppress pro-apoptotic pathways, thereby promoting cell growth and survival. The up-regulation of the expression of p21WAF-1/CIP1, a cyclin-dependent kinase inhibitor, in a dose-dependent manner by IP6 is associated with p53-mediated cell cycle arrest. p21WAF-1/CIP1 exerts its control on the cell cycle by blocking cells’ transition from the G1 phase to the S phase [61,119]. Oral administration of IP6 has been reported to augment the immune response through enhanced activity of natural killer (NK) cells [6,62] (Figure 2, Table 1). IP3, a lower inositol phosphate, produced from the degradation of IP6, appears to induce the proliferation of NK cells by promoting the intra-cellular release of free calcium (Ca^2+^) [62,125]. NK cell cytotoxicity factor (NKCF) is also released, which targets and destroys tumor cells.

Expressions of pro-apoptotic and anti-apoptotic genes are involved in cellular proliferation and apoptosis [126,127]. Bcl-2 family proteins play an important role in the induction of caspase activation and apoptosis regulation [128]. Alterations in the levels of Bax and Bcl-2 proteins may generate pro-apoptotic and anti-apoptotic activities, respectively [127,128]. Elevation of the Bax/Bcl-2 ratio is believed to be a reliable indicator of enhanced apoptotic activity [63,129]. Karmakar et al. [63] reported an increased Bax/Bcl-2 ratio in IP6 treatment and concluded that mitochondrial membrane permeability changes due to Bax elevation might explain the crucial role of IP6-mediated apoptosis. Du et al. [130] reported an elevated proportion of cell death induced by the translational product of the Bax/Bcl-2 gene. Furthermore, they observed the down-regulation of hTERT in T98G cells, which may facilitate apoptosis [63] (Table 1). hTERT is a functional catalytic protein sub-unit of telomerase that is necessary for the re-establishment of telomerase activity and thus the maintenance of genomic integrity in highly proliferative normal, immortal, and tumor cells [131]. In summary, the anti-cancer properties of IP6 present opportunities for its potential utilization in disease management. Its modulation of apoptosis provides cellular protection and prevents the subsequent development of diseases. IP6 mechanisms of action include:Free radical scavenging and mineral chelation, especially iron and calcium.Modulation of enzymes and proteins that are involved in free radical formation.

Calcium (Ca^2+^) signaling plays a significant role in a wide array of cellular processes, including those implicated in cancer cell growth, such as transcription, metabolism, proliferation, differentiation, and apoptosis [132,133]. A remodeling of intra-cellular Ca^2+^ signaling pathways may occur in cancer cells due to aberrant changes in the expression of genes related to the formation of calcium channels, pumps, and binding proteins [134,135]. This results in the de-regulation of controlled cytoplasmic increases in free Ca^2+^ concentrations, ultimately resulting in cellular proliferation and malignancy. Various anti-cancer agents have been shown to target Ca^2+^ signaling pathways and induce apoptotic and anti-proliferative processes [136,137]. Research conducted by Suzuki et al. [31], using human colorectal cancer cells, demonstrated that the lower inositol phosphates, specifically IP2-4, derived from the metabolism of IP6 by gut microbes induce intra-cellular calcium signaling and mobilization in these cells and thus could explain one modality through which IP6 expresses its anti-cancer activity.

Cytoplasmic increases in Ca^2+^ concentration occur in response to lower Ips, especially IP3, produced from diverse stimuli [138]. This IP3 binds to its receptor, a Ca^2+^ release channel, located in the membrane system of the endoplasmic reticulum (ER), resulting in Ca^2+^ efflux from the ER into the cytoplasm [138].

## 6. The Lipid-Lowering Activity of IP6

Dyslipidemia is an area of concern, since it is well-established that lipid abnormalities are directly associated with increased cardiovascular risk. Cardiovascular diseases are, in turn, the leading cause of death globally. Type 2 diabetic individuals tend to have elevated triglyceride, total cholesterol, and LDL cholesterol levels, with decreased HDL cholesterol levels. Studies have shown that IP6 has a hypolipidemic effect in rodent models, with a reported reduction in hepatic total lipids and triglycerides in animals fed myo-inositol or IP6 [139]. The proposed mechanism of action is through the inhibition of hepatic lipogenesis enzymes. Lee et al. [140] also observed a similar reduction in hepatic lipid levels, as well as a reduction in serum lipid levels in rodents fed IP6. Our studies have shown that a combination of IP6 and inositol provides a synergistic effect in the treatment of diabetes by attenuating several factors such as total cholesterol, triglycerides, intestinal amylase activity and food and fluid intake, and insulin sensitivity [140]. IP6 was also shown to positively modulate biochemical markers integrally involved in diabetes and lipid metabolism, including reducing body weight and triglycerides while increasing serum HDL in diabetic rat models [50].

Regarding food intake and metabolism, IP6 was shown to reduce leptin levels while increasing serum adiponectin [141]. This finding is significant, as leptin, a secretagogue of adipocytes, plays key roles in regulating glucose and lipid metabolism in peripheral tissues and may, therefore, indirectly modulate serum lipid concentration [142]. In addition to its effects on peripheral tissues, leptin also acts centrally by way of established leptin signaling pathways. Studies on rat models show that animals that lack the long isoform version of the leptin receptor (LepRb) in cell membranes present with severe obesity, dyslipidemia, and insulin resistance compared with leptin-deficient models [143]. A study investigating cardiac steatosis indicated that leptin is an effective lipolytic tool while reducing lipogenesis via various molecular pathways, including ATGL/HSL and SCD-1/DGAT1 pathways [144]. It is well established that low leptin levels are directly linked to weight loss and hypolipidemia. Additionally, by attenuating various molecular pathways, leptin can modify lipid metabolism.

Although leptin production is primarily in white adipose tissue, lower levels have been seen in other body tissues, such as brown adipose tissue (BAT), placenta, fetal tissue, stomach, muscles, bone marrow, teeth, and brain [145,146]. It circulates in free and protein-bound forms in the blood; however, the biologically active form is the free component [146]. Leptin bio-availability depends on the equilibrium between free and protein-bound leptin [147]. The hormone’s (leptin) ability to effectively reduce food intake and body weight led to its initial utility for treating obesity. However, obese individuals with high circulating hormone levels have demonstrated insensitivity to the exogenous administration of leptin, a phenomenon described as leptin resistance, thus limiting its clinical utility in obese individuals [148]. Of note, however, is the report of Hyun-Seuk Moon et al. [149] showing that leptin-binding protein and anti-bodies against metreleptin increased in response to metreleptin treatment, which resulted in the limitation of circulating active free leptin to about 50 ng/mL despite total leptin levels of about 982.7 ng/mL in obese diabetic individuals. Hoffmann et al. [150] demonstrated that leptin administration within the sub-physiological (0.1 and 0.5 mg/kg body weight/day) to physiological (3.0 mg/kg body weight/day) range diminished atherosclerotic disease and suggested that leptin’s anti-atherogenic effects may indirectly be through the reduction of hypercholesterolemia and liver steatosis and the up-regulation of insulin-sensitizing and athero-protective adiponectin. Thus, leptin dose-dependently improves glucose homeostasis, lipid metabolism, and liver function parameters in ob/ob mice. In a human study, Chrysafi et al. [151] reported that blood leptin levels at baseline, which ranged between 1.5 and 8 ng/mL, did not correlate with the % weight loss due to leptin treatment. They suggested that the response to leptin treatment may not depend linearly on the leptin blood concentrations concerning weight loss. Perakakis et al. [152] reported that treatment with leptin could be effective in patients with certain cardiometabolic diseases associated with leptin deficiency but not in common obesity. Hence, strict thresholds in defining leptin blood concentrations as reliable predictors of weight loss with leptin treatment in obese populations may be challenging [151,152]. Recent approaches in leptin use include identifying individuals with common obesity who have low leptin levels and more room for leptin to act when its levels are raised from low (below physiological level) to normal.

Insulin has been shown to stimulate the production and secretion of leptin in adipose tissue. However, the role of leptin in the development of diabetes remains controversial. While Soderberg et al. [153] reported a positive association between high leptin levels and future development of diabetes in Mauritian men, other studies have suggested a potential therapeutic role for leptin in managing hyperglycemia and insulin resistance in animal models of type 2 diabetes [154]. Our previous study, which examined the effects of combined IP6 and inositol supplementation in type 2 diabetic rats, indicates that the notable rise in serum leptin levels could potentially contribute to the reduction of food intake and prevention of body weight gain [30]. We then hypothesized that combined IP6 and inositol supplementation might promote insulin sensitivity in type 2 diabetic rats by increasing leptin levels in the blood. However, considering the reported varied effects of leptin in human and animal studies, the role of IP6 on leptin activity in humans needs further investigation.

Adiponectin improves insulin resistance and is a key regulator of glucose and lipid metabolism and exhibits anti-inflammatory properties. The hormone can increase HDL while decreasing serum TG concentrations by improving insulin resistance and modulating lipoprotein lipase and hepatic lipase activity [144]. IP6 promotes insulin sensitivity by increasing serum adiponectin levels, leading to reduced diabetic complications, including hyperglycemia and hyperlipidemia [155]. Improving insulin resistance is also directly linked to increased lipoprotein lipase (LPL) expression. Increased LPL expression and activity increase triglyceride lipolysis in VLDL and chylomicrons, reducing the tendency towards dyslipidemia [156,157]. A review by Omoruyi et al. [36] also summarized other mechanisms involved in the hypolipidemic effects of IP6 and inositol supplementation. Some mechanisms involve the reduction of serum lipids because of increased intestinal lipase activity due to reduced body weight arising from increased leptin production [30,158]. Another mechanism of hypolipidemic effects of IP6 was highlighted in a study utilizing 3T3L-1 adipocyte cell lines. It was found that inositol and IP6 were able to increase insulin sensitivity and glucose uptake in the cells while also increasing adipogenesis but inhibiting lipolysis [53]. The mechanism by which IP6 may modulate adipogenesis and lipolysis may be in part due to the activation of nuclear receptors, specifically peroxisome proliferator activated nuclear receptors (PPARs) by IP6 [159]. While several of these nuclear receptors exist, PPAR gamma is highly expressed in adipocytes, where it plays important roles in adipocyte differentiation, adipogenesis, etc. [160]. By way of directly interacting with the by-products of metabolism, as well as modulating the activities of enzymes and hormones involved in carbohydrate and lipid metabolism, IP6 has been shown to confer direct and indirect effects on lipid metabolism, which ultimately results in hypolipidemia.

## 7. IP6 and Bone Health

Studies surrounding bone metabolism, mineralization, and loss are critical to understanding bone pathologies and potential treatment options. As such, it has become important to study the molecular mechanisms involved in the specific cells involved in the process, inclusive of osteoblasts, osteoclasts, and osteocytes. As seen in osteoporosis, specific attention is paid to osteoclasts, since uncontrolled up-regulation in their activity can lead to excessive bone turnover and generalized bone loss. IP6 was shown to be a selective inhibitor of osteoclastogenesis, which may make it a useful compound for treating osteoporosis [161] (Figure 3). This observation suggests that IP6 can be considered an anti-osteoclastic factor and may play significant roles in maintaining bone structure by reducing bone resorption. In vitro studies confirm that the compound is non-toxic to bone marrow-derived macrophages while regulating genes important in the inflammatory response by a process called macrophage polarization [54]. Wee et al. [54] proposed that IP6 significantly impacts macrophage polarization by modulating gene expression in M0 bone marrow-derived macrophages (BMDM). Macrophage polarization plays a crucial role in regulating inflammation-induced pathogenesis. Consequently, IP6 has the potential to serve as an environmental cue to beneficially modulate macrophage behavior. Macrophages possess intrinsic properties that enable them to readily adapt to surrounding stimuli and switch their cellular functions on or off by generating versatile signaling networks. Polarized macrophages can be broadly classified into classically activated pro-inflammatory M1 macrophages and alternatively activated anti-inflammatory M2 macrophages. M2 macrophages can be further subdivided into distinct sub-types (M2a, M2b, M2c, and M2d) based on environmental cues and associated transcriptional changes in cytokine production [162,163,164]. M2a macrophages contribute to tissue repair by secreting IL-10, TGFβ, Arginase 1 (ARG-1), and Early Growth Response 2 (EGR2) [165]. The M2b sub-type facilitates immunomodulation by secretion of IL-10, IL-6, TNFα, and IL-1β [164]. M2c macrophages, like M2a macrophages, secrete IL-10, TGFβ, and ARG-1 but lack EGR2 production [165]. The M2d sub-type, commonly referred to as tumor-associated macrophages (TAMs), promotes tumor progression by secreting TNFα, IL6, IL-10, TGFβ, and Vascular Endothelial Growth Factor A (VEGFA) [166]. Through IP6 supplementation, macrophages can be polarized to an M2a-like sub-type by up-regulating the gene expression of ARG-1 and EGR2. Considering that VEGFA gene up-regulation is associated with TAMs or M2d, Wee et al. [54] demonstrated that treating BMDM with IP6 did not alter the expression of the VEGFA gene. This finding suggests that IP6 is unlikely to promote cancer progression since it tends to skew macrophage polarization toward an M2a-like sub-type rather than an M2d sub-type, which promotes tumor angiogenesis via the up-regulation of the VEGFA gene expression. This process is integral in regulating the pathogenesis of bone diseases; therefore, a greater understanding of the mechanisms involved could lead to treatment options for osteoporosis and other disorders of bone degradation [167]. From a clinical perspective, the modification of bone repair via calcium phosphate cement by adding IP6 resulted in a material with better bone-remodeling characteristics, making it highly applicable for bone repair [168]. Accelerated bone loss of post-menopausal women is of interest, since there is increased vulnerability to physiological breaks and osteoporosis. Studies on women in this cohort have shown increased bone density in those who consumed diets high in IP6, which could potentially reduce the chances of bone damage [169].

A suggested protective mechanism exerted by IP6 with regards to protecting bone density may be via a physic–chemical adsorption mechanism [170]. Interestingly, although the mechanisms involved in bone metabolism differ from those of kidney stone formation, some similarity exists, especially with regard to the inorganic ions that are involved in the process. IP6 was shown to reduce calciuria in patients with hypercalciuria because of increased bone resorption, resulting in a reduced tendency for kidney stone formation in these patients [171]. The beneficial effects of IP6 with regards to bone health may be attributable to the protective role in bone decalcification by combining with hydroxyapatite, thereby reducing the rates of its dissolution [64] (Table 1). Hence, this significant finding warrants further investigation, as hydroxyapatite plays key roles in bone structure and strength. 

## 8. Clinical Assessments and Potential Adverse Effects of IP6

As mentioned earlier, IP6 is a naturally derived compound from plants; however, artificial forms of this compound are also available in the market. Both forms have been studied for their effects on various diseases, including breast cancer, type 2 diabetes mellitus, and cardiovascular calcification [172]. In a double-blind randomized clinical trial, Proietti et al. [173] investigated the effect of IP6 on 20 adult breast cancer patients. One group received 200 mg of IP6 daily, while the second group (control) received a gel containing hyaluronic acid (5 g) daily. Both groups were treated for six months. At the end of the six months, the group receiving IP6 showed improved life quality and the ability to perform daily activities. It reduced adverse effects associated with chemotherapy compared with the control group. In another study, Sanchis et al. [48], in a crossover randomized clinical trial study (Table 1), noted a decrease in the levels of advanced glycation end-products (AGEs) and HbA1c compared with baseline measurements. A double-blind randomized crossover clinical study conducted by Ikenaga et al. [174] observed a decrease in serum uric acid levels in healthy volunteers who were administered 600 mg of IP6 compared with the control group. In a case report, Khurana et al. [175] noted that a patient with metastatic melanoma who declined systemic therapy with both immunotherapy and targeted therapy but opted for combined inositol and IP6 (800 mg/220 mg) supplement (five tablets in the morning and five in the evening daily), showed significant improvement after six months and complete clinical and radiological remission after being on the combination for two years. The patient remained in remission three years later and continued to take the combination supplement daily.

Two artificial forms of IP6 (SNF472 and Fe-PaHCP) have been clinically investigated for their effects on cardiovascular calcification in hemodialysis patients and patients with various stages of kidney disease. Perelló et al. [176], in a randomized double-blind placebo-controlled clinical study with different doses of IP6 (0.5, 5, 9, and 12 mg/kg) administered through 4 h infusions, observed a non-significant dose-dependent increase in maximum SNF472 concentration measured in the plasma and area under the curve parameters. Adverse events were also noted during the treatment period; 9 mg/kg was determined as the effective dose. Calcium phosphate crystal formations that are associated with cardiovascular calcification decreased significantly in patients infused with 9 mg/kg during the dialysis compared with the control group. However, in another randomized double-blind clinical trial, hemodialysis patients were administered different doses of IP6 (SNF472)—1, 3, 5, 12.5, and 20 mg/kg—three times a week. No significant difference in maximum SNF472 concentration measured in the plasma and area under the curve parameters was observed and no adverse effects were reported. They determined 10 mg/kg as the effective dose and administered it to hemodialysis patients for four weeks. They noted a decrease in hydroxyapatite crystallization in the IP6-treated group compared with the control group. Sanchis et al. [64] showed that phytate inhibits hydroxyapatite dissolution in a concentration-dependent manner, a feature which is important in maintaining normal bone mineral density and lowering the risk on bone-degenerative diseases (Figure 1). Raggi et al. [177], in a clinical study involving hemodialysis patients, 300 mg and 600 mg of IP6 (SNF472) were administered thrice daily for 52 weeks and they noted that the group receiving 600 mg of SNF472 showed less aortic valve and coronary artery calcification than the control group. These studies demonstrated beneficial effects of IP6 in breast cancer and hemodialysis patients, as well as cardiovascular calcification, with minimal or no adverse effects. However, it is important to note that the clinical trials mentioned above had a limited number of patients and a relatively short intervention period [172]. Therefore, future clinical trials are necessary to establish the exact therapeutic dosage, treatment duration, and potential drug interactions and side effects of IP6 supplementation. However, Brehm et al. [178] conducted a study in which they found that an elevated dietary uptake of IP6 (16 µM) resulted in larger platelet aggregates. Given that patients with cancer face an augmented risk of thrombosis [179], it is plausible to suggest that high IP6 plasma concentrations could exacerbate this risk. Consequently, the use of IP6 as a dietary supplement may not be advisable for individuals with cancer. Hence, their findings need to be investigated in vivo and in clinical practice.

## 9. Conclusions

Overall, the beneficial properties of IP6 allow for its exploitation in the management of numerous diseases, including diabetes mellitus, cardiovascular diseases, bone diseases, etc. Its modulation of apoptosis and inflammatory processes provide for the protection of cells and has implications for delaying disease onset. IP6 mechanisms of action include free radical scavenging activities and mineral chelation, especially iron and calcium, as well as the modulation of enzymes and proteins that are involved in free radical formation. The anti-diabetic properties of IP6 include its ability to physically bind to products of digestion as well as modulate enzymes critical to the process, with the result being a reduced glycemic index of foods. The ability of IP6 to modulate digestive hormones, as well as enzymes in lipogenesis, has implications for adipogenesis, obesity, insulin sensitivity, and rates of hepatic lipogenesis. On a molecular level, IP6 may also control dyslipidemia by the activation of nuclear receptors. These factors highlight the importance of IP6 in modulating serum glucose and lipids, which are key goals in controlling diabetes mellitus and dyslipidemia. IP6 is considered an anti-osteoclastic factor, which has implications for the pathogenesis of degenerative bone diseases. By way of attenuating molecular processes, including macrophage polarization as well as physic–chemical mechanisms, IP6 plays key roles in the maintenance of bone structure. The cellular and molecular activities of IP6 are well-documented and hold significant implications for disease prevention. Further studies are therefore needed to fully unravel the molecular mechanisms underlining the mechanism of action of this dynamic molecule and its derivatives.

## Figures and Tables

**Figure 1 biomolecules-13-00972-f001:**
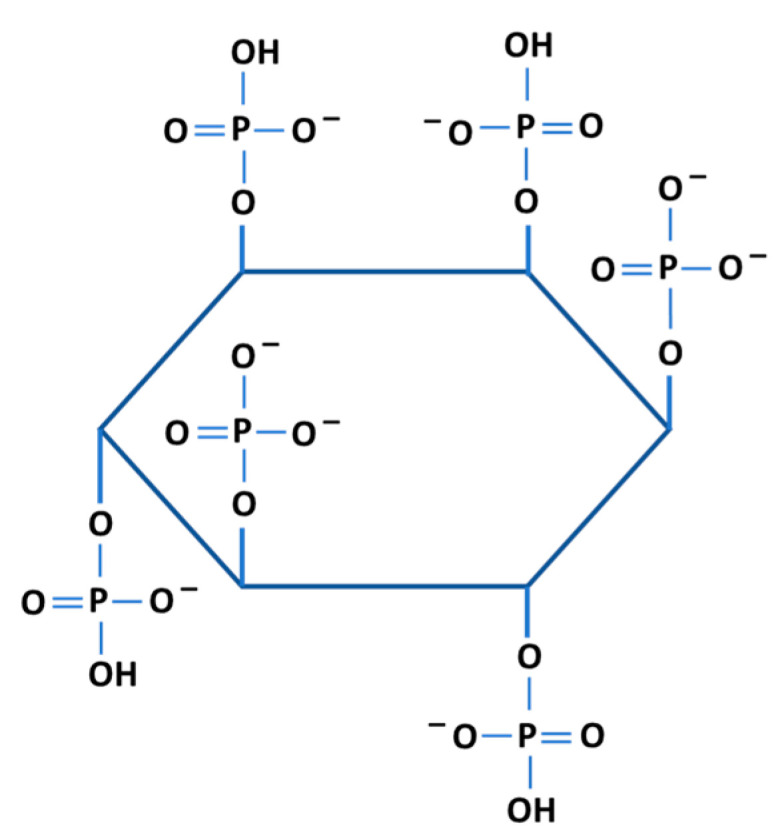
A proposed representation of the IP6 molecule comprising a hexose carbon backbone to which six phosphate groups are attached. This arrangement results in the overall electronegativity of the molecule.

**Figure 2 biomolecules-13-00972-f002:**
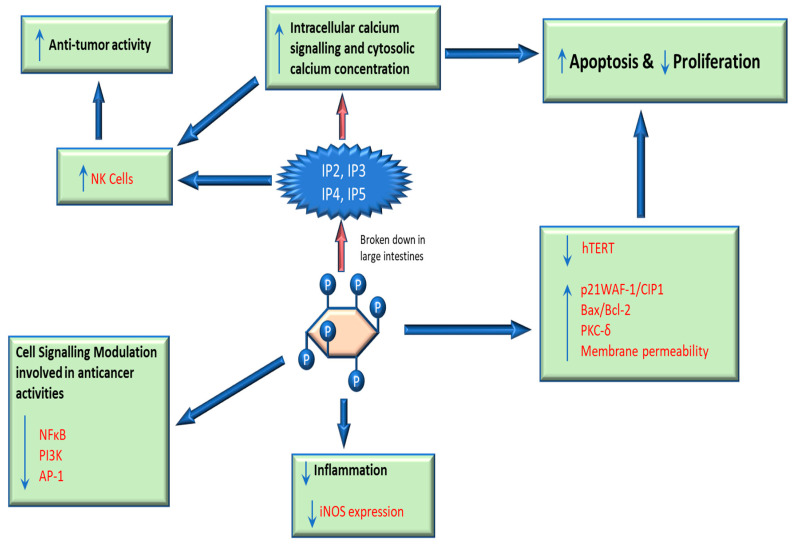
Functional attributes of IP6 and its lower derivatives on markers of biological processes implicated in development of cancers. IP6 modulates the expression of key proteins involved in biological processes associated with cancer development, such as calcium signaling, apoptosis, inflammation, proliferation, and cell signaling, while displaying immunity-boosting properties.

**Figure 3 biomolecules-13-00972-f003:**
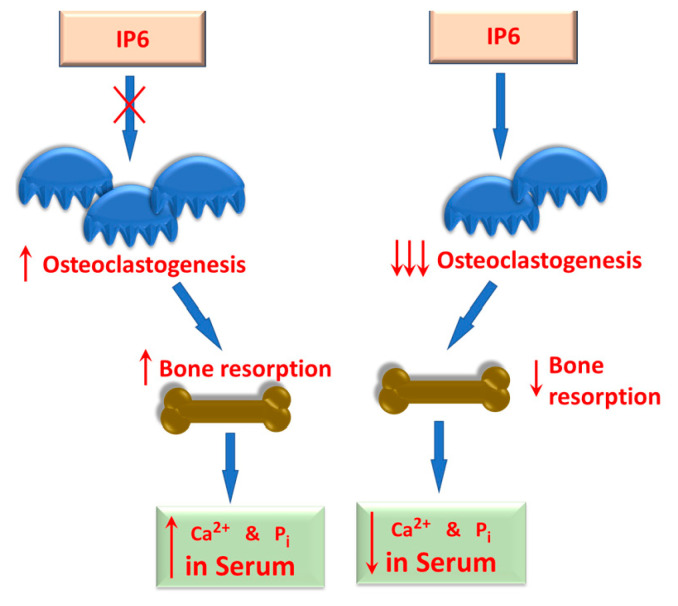
One proposed effect of IP6 on bone metabolism. IP6 can modulate the proliferation of osteoclasts (osteoclastogenesis), thereby controlling the rate of bone resorption. In the absence of IP6, the rate of osteoclastogenesis increases (as indicated by the upward arrows), resulting in increased bone resorption and increased serum calcium and inorganic phosphate. In the presence of sufficient IP6, decreased osteoclastogenesis and bone resorption (as indicated by the downward arrows) are observed as indicated by the downward arrows. This may be significant in persons who have pre-existing bone pathologies.

**Table 1 biomolecules-13-00972-t001:** Highlights of a summary of in vitro and in vivo studies showing effective doses of IP6 targeted toward various disorders.

	Reference	Concentration/Doses	Observed Effects	Benefits	Disease Prevention
1	Dilworth et al., 2005 [8] Omoruyi et al., 2013 [50]	1, 4% PA, normal, diabetic rats for 4 weeks.	Decreased random blood glucose, triglyceride, and increased HDL.	Management of glucose and levels in diabetes.	Diabetic and cardiovascular disease.
2	Foster et al., 2016 [30] Foster et al., 2017 [51] Foster et al., 2019 [35]	650 mg/kg body weight/day of PA + inositol type 2 diabetic rats for 4 weeks.	Decreased blood glucose, insulin resistance, triglycerides, cholesterol, and food intake. Increased concentration of hepatic reduced glutathione and up-regulation of hepatic superoxide dismutase and catalase.	Effective in type 2 diabetes management, improvement of renal and pancreatic function. Improved anti-oxidant status and preservation of liver cell integrity.	Metabolic disorders via the regulation of some aspects of lipid and carbohydrate metabolism. Prevention of type II diabetic complications.
3	Vucenik and Sham-Suddin, 2006 [37]	1.0–2.0 g/day prophylactic and 8–12 g/day dose of PA + inositol in cancer patients. Values were extrapolated from animal data.	Enhanced anti-cancer effects of conventional chemotherapy control of cancer metastases.	Improved quality of life and safe long-term survival of cancer patients.	Potentially enhanced cancer therapy by synergistic action with conventional chemotherapy.
4	Sanchis et al., 2018 [48]	380 mg of Ca-Mg IP6 three a day, type 2 diabetic patients for 12 weeks. IP6 concentrations (0 to 2 µM) in vitro AGE formation and observed for 7 days.	Decreased AGEs and HbA1c levels and decreased AGE formation.	Effective in reducing the development of diabetes-related diseases.	Prevention of AGE-related disorders and complications.
5	Kumar et al., 2004 [52]	150 mg PA/Kg b.w. Single treatment.	Significant protection from ulcers, decrease in gastric tissue malondialdehyde levels in ethanol-treated rats, and reductions in necrosis, erosions, congestion, and hemorrhage.	Gastro-protective effects and cytoprotection of the gastric mucosa.	Inflammation and ulcers.
6	Kim et al., 2014 [53]	3T3-L1 cells treated with IP6 or myoinositol −0, 50, or 200 μmol/L for 4 or 24 h.	Increased lipid accumulation in a dose-dependent manner. Increased insulin-stimulated glucose uptake.	Increased insulin sensitivity, inhibition of lipolysis, and improved glucose uptake.	Diabetes and its complications.
7	Wee et al., 2021 [54]	Bone marrow-derived macrophage cells were treated with 200 μM PA for 24 h.	Reduced pro-inflammatory responses and up-regulation of anti-inflammatory genes.	Shaping the function of macrophages without cytotoxicity. Resolution of inflammation.	Prevention of diseases associated with uncontrolled inflammation.
8	Lee et al., 2005 [55]	Inositol (2% *w*/*v*), IP6 (2% *w*/*v*), or a combination of both were added to the drinking water of rats for 8 weeks.	Enhanced GST activity, reduced TBARS concentration, and catalase activity.	Potential prevention of chemically induced hepatocarcinogenesis by inositol and/or IP6 supplementation.	Induction of carcinogen detoxifying enzyme (GST) and scavenging of reactive oxygen species.
9	Shan et al., 2022 [56]	1 μM, 3 μM, and 5 μM PA- in vitro cancer cells for 24, 48, and 72 h. 300 mg PA/kg b.w. Swiss albino mice for 13 days.	Induced cytotoxicity in vitro, apoptosis, and cell cycle arrest. Reduced angiogenesis and a revival of the anti-oxidant defense system.	Anti-oxidant, anti-angiogenic, and anti-tumor activities.	In vitro cytotoxic effects and in vivo angiogenic and anti-tumor effect.
10	Norazalina et al., 2010 [57]	0.2% (*w*/*v*) and 0.5% (*w*/*v*) PA in drinking water fed to rats for 8 weeks.	Reduced formation of aberrant crypt foci and reduction in the incidence and multiplicity of total tumors.	Anti-tumor and reduced pre-neoplastic legion formation.	Reduction of colon cancer risk.
11	Schröterová et al., 2010 [58]	IP6 and inositol at three concentrations: 0.2, 1, and 5 mM for 24, 48, and 72 h on cancer cell lines.	Decreased proliferation and metabolic activities of all cell lines and increased apoptosis.	Anti-proliferative and increased apoptotic.	Reduction of proliferation and up-regulation of apoptosis.
12	Vucenik et al., 1998 [59]	In vitro: 0.1–10 mM IP6 on a cancer cell line for 72 h. In vivo: 40 mg IP6/Kg in 0.1 mL PBS in mice for 2 weeks.	Suppressed tumor cell line growth and reduction in tumor size.	Rhabdomyosarcoma therapy.	Suppression of tumor.
13	Wawszczyk et al., 2012 [60]	1 and 2.5 mM IP6 for 3, 6, and 12 h on cancer cell line.	Modulation of the expression of p50 and IκBα genes in Caco-2 human colon adenocarcinoma cells. Decreases in the expression of IL-6 and IL-8.	Immuno-regulatory effects on intestinal epithelium.	Anti-inflammatory.
14	Vucenik et al., 2005 [61]	2 mM IP6 for 24 h on MCF-7 human breast cancer cells.	3.1-fold increase in the expression of anti-proliferative PKCdelta.	IP6-induced apoptosis in MCF-7 human breast cancer cells.	Possible anti-proliferative and anti-cancer activity.
15	Zhang et al., 2005 [62]	2% IP6 in the drinking water of rats for 21 weeks.	Increased blood NK cell activity. Reduced tumor size and number.	Inhibits tumor growth.	Inhibition of tumor growth and metastasis.
16	Karmakar et al., 2007 [63]	0.25, 0.5, and 1 mM IP6 for 24 h on human cancer cell line.	Decreased T98G cell viability with morphological and biochemical features of apoptosis.	IP6 promotes apoptosis in human malignant glioblastoma T98G cells.	Up-regulation of apoptosis in T98G cells.
17	Sanchis et al.,2021 [64]	Diet-rich IP6 consumption greater than 307 mg/day.	Inhibits hydroxy-apatite dissolution	Maintains normal bone mineral density.	Reduces incidences of bone degenerative diseases.

INS—inositol, GST—glutathione S-transferase, b.w.—body weight, AGE—advance glycation end-product. In the provided table (Table 1), IP6 supplementation demonstrates a diverse range of effects that cannot be universally applied. The reported therapeutic levels of IP6 differ across various disease models, which may also contribute to varying adverse effects. Therefore, it is crucial to conduct a comprehensive assessment of the optimal levels of IP6 usage in human subjects to ensure favorable outcomes in the management of different diseases.

## Data Availability

Not applicable.

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
