# Peer review of "Cellular and Molecular Activities of IP6 in Disease Prevention and Therapy"

_biomolecules, 2023, doi:10.3390/biom13060972_

Round 1

Reviewer 1 Report

l.14     health and diseases - what diseases?

l.33       of chronic diseases what diseases?

Introduction

l..44 IP6 is 44 a poly-phosphorylated inositol derivative- it is desirable to give the formula

l.51 At the molecular level - rather at the cellular level, judging by what is written next

l.55 antinutritional effects-what are those?

l.55 smaller molecules (IP4, IP3, IP2, IP1) -  It seems necessary to add a little about these molecules and their occurrence and role in humans

l.56 roles of IP6 in human health and l. 57 its functioning as a regulator of gene expression -- where it is said that it is synthesized in the human body?

l. 57 IP3 containing and l. 59 IP4 containing it is necessary to explain where these compounds come from and their connection with IP6

l. 77 IP6  is also believed to form a complex with proteins that may lead to the modulation of 78 digestive enzymes in the gastrointestinal tract, which may/could have an overall impact 79 on nutrient absorption [16,31- along with that. what was not explained above about the anti-nutritional property of IP6, this proposal does not contribute to understanding how the unfavorable properties of IP6 correlate with the favorable ones - everything seems to be in the same row

l. 80 regulates – rather, modulates

l. 82 and prevention of  tumor progression - this has already been mentioned at the beginning of the introduction.

l. 83 IP6 exerts its differentiating activity via dephosphorylated 82 derivatives (IP4, IP5 -  specify, it acts exclusively through derivatives?

l. 87 IP6 has numerous properties that confer anti-diabetic, anti-neo-87 plastic, anti-inflammatory, and hypolipemic attributes to the molecule - this has already been mentioned at the beginning of the introduction.

Anti-diabetic Activity of IP6

l. 105 IP6 dephosphorylation results in lower inositol phosphates (IP1-IP5).- it is better to talk about this in the introduction, and explain which of the derivatives are synthesized in the human body, regardless of the administration of IP6

l. 106 inositol phosphates may all be involved in antidiabetic activities -  this has already been mentioned

l. 123 combined inositol and phytic supplementation - why use this combination?

l. 136 dietary consumption of IP6 with improved insulin sensi-136 tivity and reduced blood glucose levels -  doses should be given ,

l. 138 IP6 improves blood glucose levels-doses? models?

Anti-inflammatory activity of IP6

L 149 The activation of inflammatory cells, such as fibroblasts, endothelial cells, tissue macro-149 phages, and mast cells, induces inflammation-not necessary

l.154 The overexpression  of the inflammatory response-not inflammatoty, but immune

l.158 The immune system responds to injury or infection 158 by inflammation,- this is not accurate, the authors equate immunity and inflammation

l. 165 cytokines and interleukin expression-interleukins are also cytokines

l.167 Oxidative stress involvement in the pathogenesis of diabetes-authors need to explain why they are reverting back to diabetes

l.167 Oxidative stress involvement in the pathogenesis of diabetes includes:

• The loss of pancreatic β-cells.- these are all the consequences of  stress, and not its stages

l.175   One of the genes related to inflamma-175 tion and cancer encodes inducible nitric oxide synthase - the next chapter is devoted to antioxidant properties, the previous chapter is devoted to diabetes, and here everything  is somehow mixed without any transitions

l.176 Inducible nitric oxide 176 synthase catalyzes the synthesis of nitric oxide (NO) from the oxidative- it may be worth bringing these mechanisms to the beginning of the chapter, explaining where these mediators come from during inflammation

l.184 in the production of cytokines IL-1, IL-4, 184 IL-6, IL-10, and TNF-α [69]- The authors of the cited article include more of them in the group of chronic and acute cytokines than indicated by the authors of the review. Therefore, it is necessary to formulate this more accurately.

l.192 substances associated with liver damage by promoting liver fibrosis- no transition to liver disease

l.211 degradation of IκB  proteins during NF-κB induction activates the associated PKAc and the phosphorylation 2 of the p65, a subunit of NF-κB for the efficient transcriptional activation by NF-κB -. it is not clear why this proposal appears after it has been described. that NF-κB has already translocated to the nucleus

Antioxidant activity of IP6

l.229 oxidative stress, there is an imbalance between oxidants and antioxidants that results in 229 the generation of free radical species- inaccurate formulation, ROS are generated  in any case, their ratio is important

l.241 catalyzes the metabolism - it's not metabolism

l.251 Similarly-?

l.258 in extracellular tissues-?

l.261 and other phenolic compounds [82].- not all antioxidants listed before have phenolic structures

l.266 including chelation of 266 ferric ions, resistance to oxidation and apoptosis, improved immunity, and anti-inflam-267 matory activity [87-88]- This is just a repetition of what was already said in the introduction.

Figure 1: Unfortunately, in this diagram,  it is not clear how oxidative stress correlates with inflammation and immunity. All this contributes to some fragmentary idea of the action of the substance. May be useful to include signal pathways in the scheme. This needs to be shortly discussed at the end of this chapter.

Anti-cancer and Molecular activities IP6 -strange combination of different categories

l.292 phosphatidylinositol-3 kinase (PI3K),- repetition of the decoding of the abbreviation, despite the fact that many others are not deciphered at all

l. 293 protecting cells from carcinogenesis and preventing tumor angiogenesis - explain the role of the enzyme here, why its inhibition is so affected here.

l. 295 p21WAF-1/CIP1- what is it?

l.297 Oral administration of IP6 has been reported to augment the immune response 297 through enhanced activity of natural killer (NK) cells [6,106]- here, too, this effect should be discussed, since the activation of lymphocytes is associated with an increase in the activity of NFκB .

l. 301 Pro-apoptotic and anti-301 apoptotic genes are involved in cellular proliferation and apoptosis [108-109].- the meaning of this sentence after the previous one is not very clear the meaning of this sentence after the previous one is not very clear

l.309 Overall, the antioxidant and anticancer properties of IP6 allow for its 309 exploitation in the management of many diseases. you can’t just combine these two properties as inseparable; in some cases, antioxidant properties can have a pro-cancerous effect. That is why it is important to indicate not just the effect, but the conditions for obtaining it: dose, time, cell or animal line, human)

l.314 Especially iron and calcium -?

Figure 2:, the relationship of effects is not clear How is the decrease in the synthesis of TNF associated with the increase in apoptosis? Cchanges in receptor expression are not enough to promote apoptosis. Calcium concentration increases through lower phosphates, and phytin itself chelates calcium. This is not reflected in the diagram and is not discussed.  

l.339 Diabetic dyslipidemia - Dyslipidemia isn't just for diabetes

l.345 Lee et al. [120] the numbering in the text and the list does not match, check all links

l.367 These mechanisms partially explain the ability of IP6  to reduce serum lipids. -here it is desirable to give details about the use of leptin (dose, object) in order to understand where is the inhibition of leptin production, and where is the decrease to optimal values, otherwise it is not clear why the decrease in leptin has opposite effects in different cases

l. 393 and dyslipidemia- What exactly? In general, dysliidemia was not mentioned here, and diabetes is just one of the diseases associated with it.

IP6 and Bone Health

l.423 metabolism of bone and renal stones- what is kidney stone metabolism?

Figure 3: in fig. it is not clear what polarization is (although this is exactly what the figure in the text refers to) and how the drug specifically acts on it.  And in fig., the effects seem opposite of those described in the text.

Conclusions

l. 431 Overall, the antioxidant and anticancer properties of IP6 allow for its exploitation in the 431 management of many diseases.- not clearly formulated - anticarcinogenic properties - against cancer, but antioxidant - in what diseases

l. 438 The ability of IP6 and inositol-Why is  inositol here?

l. 442 IP6 in regulating serum –rather modulation

l. 448 disease prevention – what diseases?

About 52% of references up to 2010 inclusive.

a

Author Response

The paper was modified as suggested by the reviewer.

Please see attached file for individual responses to comments.

Reviewer 2 Report

This is a routine survey of the literature of the possible roles of phytic acid in biomedicine and disease prevention. It is straightforward and of suitable breadth.

Author Response

Reviewers comments are duly noted.

Reviewer 3 Report

In this paper the authors have reviewed the information on IP6 activities in disease prevention.

The paper is interesting although it has several shortcomings.

First of all, the review includes studies carried out in vitro and in vivo, the latter carried out both with experimental animals and clinical studies with humans. However, in the article they have not been considered separately.

On the other hand, the IP6 concentration values used in cell cultures are not indicated, nor are the doses administered to animals or humans.

It should be considered that in vitro studies should be carried out at concentrations achievable in vivo.

It is recommended to include a table in the paper indicating the type of study as well as the concentrations or doses of IP6 studied in relation to the different beneficial effects of phytate.

Lastly, an assessment by the authors of the studies and the observed effects and their extrapolation to the prevention of diseases is lacking.

For this reason, it is considered that the paper cannot be published in its present form, and requires major changes.

Author Response

The authors agree with the comments made by the reviewer.

Please see Table 1 added to the article (page 14) that addresses both major concerns of the reviewer.  

Round 2

Reviewer 1 Report

1. Authors should carefully check the entire text, many repetitions, especially
in relation to IP6 derivatives (for example, l.14-l18, l31-33, l 146, l.190, l.211).

2. l.47 –here you need a graphic formula (Fig.).

3. There are many imprecise formulations (ll. 49-51, l.125-127 (physiological role?), l. 318 (humoral inflammation?), l. 400, 495, 509-510).

4 No discussion l. 200.

5, The chapter “Antioxidant activity” basically talks only about the antioxidant system, with unnecessary details that are not directly related to the action of IP6. At the same time, only 1 sentence is devoted directly to the action of the molecule. Ll. 442-444 describe a non-antioxidant action, although related to an antioxidant, but this is not described. Fig. 1 does not give any idea of ​​the relationship between the various effects of IP6, although they are given in the text, and authors have been suggested to improve the figure. Since the picture is not informative, it is better to remove it.

6. Abbreviated names of molecules are not deciphered at all (for example, p21WAF-1 / CIP1) or not at the first mention (for example, MAPK, l. 460) / The text must be checked.

7. l. 459- It is unclear why a decrease in the expression 458 of tumor necrosis factor-α (TNF-α) is accompanied by apoptosis activation, usually TNF is considered to be an apoptosis stimulator. This is not discussed here. Moreover, this is not shown in Fig. 2.

8. When describing the action of IP6 on various signaling molecules, the role of these molecules in the described processes (for example, PI3K, hTERT) is not indicated, so the conclusions on the effects are not clear.

9. l.476  it?

10. Appearance in the caption to fig. and in a table of data missing from the text itself.

So, in fig. 2 such text appears (l.550). And if it comes to that, then in fig. and in the legend it is necessary to indicate what is ultimately affected by the change in Ca concentration. In addition, fig. 2 does not indicate communication between processes, although the text says so. Moreover, an increase in immunity and a decrease in inflammation are simultaneously indicated. These are, as it were, opposite processes, and influence on them is important at different stages of cancer development, so it needs to be pointed in the text and fig.

 11. The text is diluted with unnecessary information regarding leptin (l. 585-619).

12. There is a hint of involvement of macrophage polarization in the mechanisms of action of IP6 (l, 671), but data on the direction of this action of IP6 have not been presented

13. There is no reference in the text to Table.

14. In the first review, the authors were asked, in cases of conflicting results, to discuss this by presenting doses and subjects of study. The authors did not do this, but simply presented the data in the table, without discussion. At the same time, data appeared in the table that were not in the text. Their appearance should be somehow stipulated in the text, where the table should be mentioned. The table itself is made inaccurately. So, p. 3 - the conclusion about "prevention" is made. But this effect does not exist. Point 8 - columns 3 and 4 are mixed up? When mentioning the anti-cancer effect, it is desirable, as in the text, to indicate the type of cancer or cell line. However, in most cases, the authors have not described the anticancer effect, but only highlighted the effect on the mechanisms that may be involved in the development of cancer. Accordingly, the authors can only talk about the potential  anti-cancer effect, which must be indicated in the column caption.

No effect on the bones in the list of effects.

15. It is not discussed why IP6 is more effective in the company of inositol.

16. It is still not clear how the effect of IP6 is manifested in clinical studies in terms of side effects. Although the authors focused only on the positives, briefly pointing out the need for further research to assess the negative impacts, some data on this topic appeared in the literature, but the authors cite little data from the last 5 years.

Reviewer 3 Report

I believe the manuscript has been sufficiently improved to warrant publication in Biomolecules.

Author Response

Noted